# Clinical Potential of YY1-Hypoxia Axis for Vascular Normalization and to Improve Immunotherapy

**DOI:** 10.3390/cancers16030491

**Published:** 2024-01-23

**Authors:** Concetta Meo, Filomena de Nigris

**Affiliations:** Department of Precision Medicine, School of Medicine, University of Campania “Luigi Vanvitelli”, 80138 Naples, Italy; concetta.meo@studenti.unicampania.it

**Keywords:** tumor vascularization, hypoxia, immunotherapy

## Abstract

**Simple Summary:**

Solid tumors create a hostile hypoxic microenvironment characterized by abnormal vascularization and immune suppression. Target tumor vessels and the activation of immunostimulatory cells have shown promising results in suppressing tumor growth, leading to the development of a variety of strategies. In this review, we dissect the concept of vessel and immune normalization and discuss the YY1-Hypoxia axis as a potential target.

**Abstract:**

Abnormal vasculature in solid tumors causes poor blood perfusion, hypoxia, low pH, and immune evasion. It also shapes the tumor microenvironment and affects response to immunotherapy. The combination of antiangiogenic therapy and immunotherapy has emerged as a promising approach to normalize vasculature and unlock the full potential of immunotherapy. However, the unpredictable and redundant mechanisms of vascularization and immune suppression triggered by tumor-specific hypoxic microenvironments indicate that such combination therapies need to be further evaluated to improve patient outcomes. Here, we provide an overview of the interplay between tumor angiogenesis and immune modulation and review the function and mechanism of the YY1-HIF axis that regulates the vascular and immune tumor microenvironment. Furthermore, we discuss the potential of targeting YY1 and other strategies, such as nanocarrier delivery systems and engineered immune cells (CAR-T), to normalize tumor vascularization and re-establish an immune-permissive microenvironment to enhance the efficacy of cancer therapy.

## 1. Introduction

In solid tumors, a hypoxic microenvironment is one of the most common pathogenic conditions that promotes the formation of new vessels and favors immune escape [1]. Neovascularization is considered a milestone in tumor growth as it is the key mechanism supporting the proliferation of cancer cell clones and their dissemination to distant sites. The sprouting of existing vessels is thought to be the main initial event through which angiogenic growth factors promote new vessel formation from pre-existing vessels. However, several other vasculogenic mechanisms strongly associated with tumor hypoxia have been recognized, including intussusceptions, co-option, and vasculogenic mimicry [2,3,4]. Consequently, the vasculature within tumors is heterogeneous, abnormal, dysfunctional, and hyperpermeable. This condition determines insufficient nutrient and oxygen perfusion in selected tumor areas, such as the core far from the vessel, or in the high-proliferation cohort of cells. Under reduced oxygen tension, tumor cells adapt to hypoxia and activate several survival pathways, including hypoxia factor 1α (HIF-1α) and vascular endothelial growth factor (VEGF), able to generate a vicious cycle with immune cells.

VEGF, a key driver of angiogenesis, has also been demonstrated to be an immunosuppressive factor, interfering with the maturation of dendritic cells (DC) and suppressing T cell priming [5]. Moreover, cell components of the hypoxic tumor microenvironment (TME) increase chemokine secretion, including stromal cell-derived factor 1-α (SDF1-α) and C-C Motif Chemokine Ligand 28 (CCL28), and recruit regulatory T helper cells (Tregs), a heterogeneous population of immature myeloid cells (MDSCs), and M2-type macrophages. These immune cells contribute to an immunosuppressive microenvironment and promote abnormal vessel growth. In particular, M2-like macrophages secrete pro-angiogenic factors, mainly VEGF, which promote the development of excessive and immature vessels [6,7]. Therefore, such a hostile microenvironment also limits CD8^+^ T and CD4^+^ Th1 cell infiltration [8,9,10] (Figure 1). Consequently, strategies to induce a TME permissive to immune cell infiltration are of major interest to improve the efficacy of immune therapy and reduce tumor growth. However, therapeutic targeting of vascularization mechanisms has proven to be inefficient and, in some instances, exacerbates tumor progression [11,12]. This unexpected outcome, together with better knowledge of the contribution of immune cells to tumor vascularization mechanisms, has led to the concept of vascular normalization [13]. Its goal is to modulate abnormal vascular development, favoring blood perfusion within the tumor and opening the way to immunotherapy and other drugs [14,15]. Recent research has used various therapeutic approaches, ranging from optimal doses of antiangiogenic drugs alone or in combination with immune checkpoint inhibitors (ICIs) [16]. However, data emerging from trials have revealed the poor efficacy of this treatment because multiple mechanisms of immune suppression and vasculature heterogeneity, independent of VEGF, coexist in tumors. For clinical applications, additional exploration of hypoxia-response pathways is warranted to optimize vascular normalization, improve antitumor immunity, and avoid tumor resistance. In the following section, we describe the current vascular and immune normalization strategies and highlight the clinical potential of the HIF-YY1 signaling axis.

## 2. Vascular and Immune Crosstalk Normalization

Antiangiogenic therapy, initially developed to eliminate tumor blood vessels, results in increased hypoxia and tumor aggressiveness [11]. Growing evidence has demonstrated a complicated interconnection between vessels and immune cells, in which pro-angiogenic molecules have an immunosuppressive role and immune cells promote the proliferation of regulatory T cells and vessel immaturity. These observations indicate the need for therapeutic strategies to normalize the tumor vascular blood supply and immune modulate the TME permitting the infiltration of activated immune effector cells [17,18].

Therefore, ICI treatment administered concomitantly with antiangiogenic drugs has been shown to be more effective than single [13,17,19]. Indeed, targeting both the immune microenvironment and tumor vessels has been approved by the FDA for the treatment of liver, kidney, lung, and uterine cancer [20]. However, the efficacy of these combinatorial regimes differs between cancer types and is not free of adverse effects [21]. Trials have indicated that the duration of treatment, sequencing of therapies, and tumor-specific hypoxia pathways influence efficacy. All these aspects need to be better evaluated to improve the treatment benefits and patient outcomes. The future of antiangiogenic regimens seems to consist of tumor-specific treatments to normalize vascular and immune-cross talk and novel immune-cell technologies [21] (Figure 2).

## 3. Vascular Signaling by YY1

### 3.1. HIF-YY1 Vascular Signaling Axis

A plethora of mechanisms triggered by hypoxia affect tumor vascularization. Hypoxia is a hallmark of the TME of various cancer types, generated by the activation and stabilization of hypoxia-inducible factor-1 (HIF-1) [22], and its signaling is well known to promote the angiogenic switch [23]. Overexpression of HIF-1α has been shown to be closely associated with poor prognosis, increasing tumor growth, vascularization, and metastasis [15]. The general mechanism of HIF-1α activity includes its heterodimerization with p300 and the formation of a transcriptional activation complex. This can recognize the HRE-DNA site and activate the transcription of target genes, such as VEGF, angiopoietin, fibroblast growth factor 2 (FGF2), and platelet-derived growth factor (PDGFB) [16]. HIF-1α also regulates genes involved in cell survival and metabolism, such as phosphoglycerate kinase (PGK), carbonic anhydrase 9 (CA9), Bcl-2 interacting protein 3 (BNIP3), and glucose transporter 1 (GLUT1), all of which contribute to tumor progression [24]. Many binding partners can stabilize HIF-1α on specific promoters of genes, including the Yin Yang 1 protein (YY1) protein. This is a multifunctional zinc finger protein, a member of the GLI-Kruppel family, and is part of the polycomb complex [25]. The expression and function of YY1 have been reported in both normal and cancerous cells [26]. YY1 may activate or deactivate gene expression, depending on the binding partner, functional interactions with corepressors or activators, and chromatin status, resulting in different phenotypic effects [26,27,28]. YY1 is highly expressed in numerous cancer types, and increased levels correlate with poor clinical outcomes. Increased expression of YY1 was reported in several cancer types, including prostate cancer [29], osteosarcoma [30], breast cancer [31], glioma and meningioma [32], gastrointestinal cancer [33], pancreatic ductal adenocarcinoma [34], melanoma [35], and hepatocellular carcinoma (HCC) [36]. In many types of cancer, oncogenic tumor signaling triggered by YY1 involves the activation of the c-Myc oncogene [37], AKT [38], and inhibition of tumor suppression by p53 [39,40].

YY1 also shapes the TME under hypoxia, thereby promoting neoangiogenesis. Data demonstrated that YY1 cooperates with HIF-1α as a binding partner on the VEGF promoter, stabilizing HIF-1 transcriptional activity and increasing VEGF expression [41,42,43,44].

YY1 also regulates HIF-1α post-translationally because their interaction prevents proteasomal degradation of HIF-1α in hypoxic conditions [45]. This was confirmed by the observation that under hypoxia, YY1 knockdown disrupts the stability of HIF-1α and decreases the expression of target genes (e.g., VEGF, Transforming Growth Factor (TGF-α), PGK, CA9, BNIP3, and GLUT1) in a p53-independent manner [43]. Consistent with this, YY1 silencing impairing tumor vascularization reduced cancer growth and metastatic lung colonization in p53-deficient tumors [43]. Indeed, it has been demonstrated that p53 is negatively regulated by YY1, which facilitates direct binding to E3 ligase mouse double minute 2 (MDM2), thereby enhancing its ubiquitination and proteasomal degradation [39].

In tumor hypoxia, many chemokines can be released, including C-X-C motif chemokine 4 (CXCR4), which affects angiogenesis, immune surveillance, and the ability of tumor cells to metastasize [46]. Studies on osteosarcoma have shown that downregulation of YY1 negatively regulates the VEGF/CXCR4 axis pathway, thereby inhibiting angiogenesis and tumor cell migration by reducing the transcription and activity of MMPs [41,42]. These findings suggest that YY1 is a target in tumor therapy independent of p53 status. This is important because p53 is frequently mutated or downregulated in tumor cells. Figure 3 illustrates the angiogenic mechanism mediated by YY1.

### 3.2. YY1 Promotes Tumor Angiogenesis in HIF-1α-Independent Manners

YY1 may promote tumor angiogenesis in a HIF-1-independent manner via the DEK oncogene [47]. The oncoprotein DEK, a non-histone chromosomal factor, is overexpressed in various cancers, such as lung and prostate cancers [48,49], and is correlated with poor clinical outcomes [50]. DEK has been implicated in multiple cellular processes, including transcriptional regulation, mRNA processing, chromatin remodeling [51,52], cell proliferation, differentiation, and apoptosis [53,54]. It was originally identified as a fusion protein with the CAN nucleoporin in a subtype of acute myeloid leukemias [55]. Zhang et al. demonstrated a novel function of DEK following transcriptional activation via YY1 [47]. In the model proposed, DEK enhanced VEGF transcription by directly binding its responsive element (DRE) to the VEGF promoter or indirectly by stabilizing HIF-1α [56]. The authors demonstrated that DEK physically interacts with HIF-1α and p300, forming a complex with the VEGF promoter in breast cancer cells [47].

Tumor angiogenic mechanisms can be triggered also by proinflammatory cytokines (such as IL-8) [57], chemokines such as CXC motif chemokine ligand 1 (CXCL1) [58], or thrombospondin-1 (TSP-1) and TSP-2 [59] via sonic hedgehog (Shh) signaling. In lung cancer carrying Kras mutations, it has been demonstrated that the YY1 transcription factor is a crucial mediator between the mutated isoform of Kras and Shh signaling [60]. When Kras mutations were present, YY1 promoted neovascularization by increasing the ZNF322A expression that activated Shh pro-angiogenic signaling. The same group identified the Kras/YY1/ZNF322A/Shh transcriptional axis as part of an important mechanism underlying neo-angiogenesis and migration of lung cancer cells in in vitro/vivo models [60].

YY1 in endothelial cells (EC) is important for vascular development and angiogenesis. Its knockout determines vascular deficiency and embryonal lethality. Mechanistically, YY1 functions as a modulator protein of Notch signaling, controlling vascular development and EC functions, such as proliferation, migration, lumen formation, and vessel stability [61,62]. In particular, YY1 directly interacts with the N-terminal domain of the recombination signal binding protein for the Ig Kappa J region (RBPJ), competing with the Notch coactivator mastermind-like protein 1 (MAML1) for binding to RBPJ, and thereby impairs formation of the NICD/MAML1/RBPJ complex [63]. YY1 could therefore be a molecular target for the treatment of angiogenesis-related diseases (Figure 3).

### 3.3. YY1-Related Non-Coding RNAs (ncRNAs) in Angiogenic Mechanisms

In malignancies and under chronic hypoxia, YY1 also acts as the critical interface between epigenetic code and noncoding-RNA regulatory mechanism [45].

In some tumors, e.g., acute leukemia, YY1 acting as a transcription factor promotes the overexpression of small nucleolar RNA host gene 5 (SNHG5), which complements and binds miR-26b and reduces angiogenesis [64]. Reciprocal regulation between YY1 and miRNAs has also been reported. In acute myeloid leukemia, YY1 negatively regulates let-7a, which inhibits the expression of the anti-apoptotic protein B-cell lymphoma-extra-large (BCL-xL) [65]; instead, miR-7 may silence YY1 and KLF4 mRNAs, contributing to chemoresistance [66]. Several classes of RNAs, such as long non-coding RNA and circ-RNAs, are under the transcriptional control of YY1. In lung cancer, YY1 overexpresses the non-coding RNA MCM3AP-AS1, which promotes cancer progression by binding to and degradation of miR-340-5p, a negative regulator of angiogenesis [67]. However, YY1 can also be regulated by circ-RNAs. In particular, in cholangiocarcinoma (CCA), circ-CCAC1 promotes CCA progression and angiogenesis by degradation (sponging) miR-514a-5p, a negative regulator of YY1. Collectively, these data indicate that the circ-CCAC1/miR-514a-5p/YY1/CAMLG axis plays an important role in angiogenesis [68] (Figure 3).

Thus, many studies have confirmed that YY1 and non-coding RNAs are involved in a complex crosstalk that influences tumor progression. However, these mechanisms need to be further elucidated in order to fully understand their role and importance in malignance.

## 4. Immune Cells Regulate Tumor Angiogenesis

A complex regulatory network between the TME and innate and adaptive immune cells influences the pro and antiangiogenic phenotype of immune cells [10], in particular tumor-associated macrophages (TAM). Macrophages with the M1 phenotype suppress endothelial sprouting and can induce vessel maturation by secreting TNFα [69]. In contrast, macrophages with the M2 phenotype and TAMs linking angiopoietin-2 promote angiogenesis [70]. Consistently, the knockout of macrophages expressing Tie-2 (the angiopoietin receptor) induces normalization of the tumor vasculature [71]. CD11^+^ immune-suppressing myeloid-derived cells from the macrophage lineage may promote angiogenesis, forming vessel-like tubuli containing CD31 antigens (endothelial cell phenotype) or by their integration into the tumor vasculature [72]. Adaptive immune cells and CD8^+^ T cells also influence the EC characteristics and myeloid cell phenotypes. In particular, CD8^+^ cells secreting IFN-γ inhibit the proliferation and migration of ECs [73], stimulate vascular maturation by recruiting pericytes [73], and reprogram TAMs from the M2- to M1-like phenotype [74]. In addition, CD4^+^ T helper 1 (TH1) cells promote vessel normalization by secreting additional IFN-γ. Indeed, selective depletion of CD4^+^ TH1 cells increased vessel malformation, whereas CD4^+^ T cell activation improved vessel normalization [75]. However, in many solid tumors, hypoxic conditions result in ineffective vascularization and the expression of immune inhibitory antigens, determining resistance to ICI therapy [76]. In particular, the activation of HIF-1α stimulates both VEGF-A and inhibitory checkpoints, such as the PD-1/PDL1 pathways, which favor immunosuppressive phenotypes [77]. Additionally, hypoxia factor in fibroblast-associated cancer cell populations (CAFs) positively regulates the expression of transforming growth factor-beta (TGF-β), which promotes the differentiation and proliferation of Tregs with pro-tumorigenic myeloid cell phenotypes [78].

Normalization of immune cell profiles within the TME is a promising aim of immunotherapy.

## 5. YY1 in B and T Cells

A correlation between YY1 and the immunosuppressive TME has been reported in several tumor types [27,34]. Moreover, studies have reported YY1 as a regulator of B and T cell development, activation, differentiation, and immune function [79,80] (Figure 3).

B cell development and function depend on the YY1 downstream transcriptional program [80,81,82]. Deletion of the YY1 gene prevents the transition of pro-B cells into pre-B cells at early developmental stages [79,80].

Furthermore, YY1 was identified as a controller of nuclear deaminase protein levels (AID), an enzyme required for class switch recombination (CSR) and somatic hypermutation (SHM), the processes responsible for antibody maturation, and the expression of different immunoglobulin isotypes [80,81,83]. Increased levels of YY1 are correlated with poor survival prognosis in patients with diffuse large B-cell lymphoma (DLBCL) [83,84], high-grade DLBCL, or Burkitt’s lymphoma [85], suggesting an oncogenic function of YY1 in human B-cell lymphoma genesis [86]. Indeed, YY1 binds to the Kruppel-like factor 4 (KLF4) promoter, inducing proliferation and differentiation of B-cell neoplasm in B-NHL [87].

Notch signaling is an important pathway regulating binary cell-fate choices at crucial checkpoints, including T-cell- versus B-cell-specific gene expression, αβ versus γδ T-cell-receptor expression, and CD4^+^ versus CD8^+^ lineage decisions. During physiological T cell development, YY1 has been shown to regulate Notch1 expression by binding to the enhancer locus to promote signaling in T cells. YY1-deficient T cells have reduced Notch1 expression and signaling [88]. However, in the context of an immune suppressive tumor microenvironment, YY1 regulates the proliferation and function of Tregs through remodeling of the Foxp3 locus. Targeting YY1 could be useful to modulate Treg cell proliferation and the beneficial functions of CD8^+^ T cells [89].

On the other hand, YY1 seems to promote T cell exhaustion, a phenomenon that affects CD8^+^ T cells, in which persistent antigenic stimulation renders the cells hyporesponsive and incapable of eliminating tumor cells. Prolonged T cell activation upregulates both YY1 and EZH2 proteins that repress IL-2, a cytokine involved in their capability to kill tumor cells [90]. Another mechanism by which YY1 is involved is tumor resistance to ICI immunotherapy, achieved through the positive regulation of programmed death receptor-1 (PD-1)/and LAG3. The mechanism describes the activation of CD8^+^ T cells expressing PD-1 [91] and the binding to the ligand PD-L1 on different cells of the TME, promoting survival pathway instead of apoptosis [92,93]. The presence of PD-1-positive T cells and YY1 protein was correlated with disease progression in melanoma [90,94] and immune resistance to ICIs in lung adenocarcinoma [95]. Therefore, YY1 regulates several signaling pathways, including the inhibition of p53 and miR34a, the activation of the PI3K/Akt/mTOR pathway, as well as c-Myc and COX-2 [96] involved in tumor resistance to ICIs.

## 6. Preclinical Model and Clinical Findings of Vascular Normalization and Immune Modulation

Combinations of immune and antiangiogenic therapies have been tested in several preclinical models. In pancreatic neuroendocrine tumor (RT2-PNET), mammary carcinoma (MMTV-PyMT), and glioblastoma (NFpp10-GBM) [97] mouse models, therapy based on anti-VEGFR2 and anti-PD-L1 normalized tumor vessels, favoring the intertumoral infiltration of activated T cells. On the other hand, antiangiogenic therapy was also able to reduce the TOX-mediated T cell exhaustion program in the TME, influencing the proliferation of cytotoxic T lymphocytes [98].

The most successful preclinical results have been reported in renal cell carcinoma (RCC) and hepatic cell carcinoma (HCC), which showed improved anticancer immunity and overall survival in a mouse model (OS) [99,100]. Additional promising preclinical efficacy was also demonstrated by the combination of ANGPT2/Tie2 inhibitors with anti-VEGF and anti-PD-1 therapy in both genetically engineered and transplant tumor mouse models [101]. Recently, encouraging data were provided by a multicenter, open-label, multicohort phase Ib/II KEYNOTE-146 trial (NCT02501096) [102], which reported an increase ORR of 39.6–45.3% at 24 and 48 weeks, respectively, after lenvatinib and pembrolizumab (tyrosine kinase inhibitor and anti-PD-1, respectively) treatment in endometrial cancer. In addition to the above-mentioned trials, numerous studies on various malignancies have demonstrated the efficacy of combining PD-1/PD-L1 inhibitors and anti-VEGF agents [16,103]. Nonetheless, some challenges remain, as the effective dose, timing, and duration of treatment need to be established, but studies evaluating the effects of different strategies are beginning to emerge [104].

## 7. Novel Strategies: Targeted Therapy

An alternative strategy to normalize tumor vessels and improve immunotherapy efficacy is based on novel systems to deliver nitric oxide (NO), which regulates angiogenesis, using nanocarriers as a vehicle [105]. The NO approach is of great value not only because NO is a physiological molecule but also because nitrosylation is a regulative post-translational modification of proteins. In particular, NO may inhibit YY1 function. Indeed, DETA-NONOate, a NO donor inducing S-nitrosylation of YY1, inhibits its DNA-binding activity and triggers Fas to induce apoptosis [106,107]. DETA-NONOate is still in the early stages of clinical development, but its combination with cisplatin-induced apoptosis in otherwise therapy-resistant prostate cancer cell lines [106]. Additionally, NO donors could be useful in improving the efficacy of immunotherapy by reducing YY1 activity and Treg proliferation.

Hypoxia is an integral part of the TME, and HIF-1 pathways play a central role in tumor growth by affecting multiple interconnected mechanisms, rendering a challenge to target HIF-1. In this context, given the central role of YY1 in the stability of HIF-1α on the VEGF promoter, it is conceivable that YY1 targeting might offer an innovative therapeutic approach to normalize vascularization [108]. In this regard, it is tempting to speculate that targeting tumor YY1 may normalize tumor perfusion, favoring the recruitment of effector T cells and antitumoral immunotherapy. Given that it is possible to directly target hypoxia by using HIF-1 inhibitors, one can further stimulate the production of pro-angiogenic factors and cytokines to foster vascular endothelial cell proliferation [109]. Destabilizing HIF binding and reducing YY1 expression could be alternative possibilities [109,110,111,112].

Another approach to reduce YY1 mRNA and target tumor angiogenesis could be the use of RNA interference. As demonstrated in lung tumors, MiR-29a downregulates DNA methyltransferases (DNMT) 3A and 3B and suppresses YY1 mRNA [113,114]. MiR-186 reduces YY1 expression in lung and prostate tumor cells by binding to sequences at the 3′ UTR region of YY1 mRNA, leading to decreased cell migration and invasion [115]. Several other miRNAs, as well as regulation mediated by lncRNA [116], can target YY1 in various cancer types; however, although promising results have been obtained from preclinical studies, there are currently no clinical trials.

Preclinical studies using gene editing techniques have recently yielded promising results by inhibiting YY1. Xu et al. used CRISPR/Cas9 to downregulate YY1 in prostate cancer. Lowering YY1 expression reduced tumor cell metabolism and promoted apoptosis of prostate cancer cells [117]. Above all, YY1-targeted therapy has good potential in combination with immunotherapy to improve patient response rates and resistance compared to immunotherapy alone.

An alternative approach to improve the immune response is to target the immune microenvironment. Recent advances have been achieved by CAR T cell therapy [118]. This approach involves: 1-the the molecular modification of human T cells, introducing synthetic receptors capable of recognizing selected antigens as VEGF receptor 2 or tumor-associated antigens (TAAs), 2-autologous infusion of CAR-T cells in patients where they can recognize and kill cells expressing selected antigens [119,120]. This personalized approach could address the heterogeneity of vessel origin targeting with CAR T cell antigen-specific cells. Additionally, the improvement in vessel perfusion instead of vessel destruction could be helpful in recruiting a stable immune response against the tumor. CAR T-cell therapy has shown promising results in lymphoma, inducing durable antitumor immune responses, but has exhibited low efficacy in solid tumors due to limited capability to infiltrate the TME [121]. Nevertheless, considering the overall advantages and effectiveness of CAR T-cell therapy, many studies have evaluated their use in combination with vascular normalization treatment [122], as reported in Table 1.

## 8. Conclusions

Resistance to antiangiogenic therapies is mainly attributable to the complex interactions between vessels, immune cells, and cellular components of the microenvironment that allow tumor growth [134]. It has been proven that HIF-1 and protein kinase B (AKT) are capable of driving tumor growth despite mTOR and VEGF inhibition [135]. HIF-1-driven neovascularization can also be initiated independently of VEGF and thus be resistant to angiogenesis inhibitors [136]. One mechanism of VEGF tyrosine kinase inhibitor resistance is through the direct selection of cell subpopulations that can rapidly upregulate alternative pro-angiogenic pathways [137]. Several studies in RCCs and HCCs demonstrated that following treatment with tyrosine kinase inhibitors such as sunitinib [138] and sorafenib [139], these tumors increase the release of pro-angiogenic cytokines to foster vascular endothelial cell proliferation [140].

In addition, tumor hypoxia, generated by blocking angiogenesis, activates resistance to additional target receptors, such as epidermal growth factor receptor (EGFR) inhibitors [141] in non-small cell lung cancer and in a pancreatic cancer model [140]. With regard to the immune normalization of the TME, recent data from single-cell omics have demonstrated their diversity. Therefore, it is not surprising that immune cells showed different functions in different tumor types. However, as mentioned above, targeting vessels influences immune components and vice versa. Strategies combining vessel and immune normalization may yield greater efficacy, and studies are currently evaluating the effects of different combinations and dosages. In this context, YY1 may be an opportunity to prevent HIF function, inducing longer-term vascular normalization and enhancing the trafficking of T cells and immunotherapy efficiency. However, a better exploration of YY1-hypoxia-signaling pathways for clinical application, particularly in solid tumors, will be essential [142]. Nevertheless, YY1-targeting is a promising antitumor strategy, particularly in combination with other treatments.

## Figures and Tables

**Figure 1 cancers-16-00491-f001:**
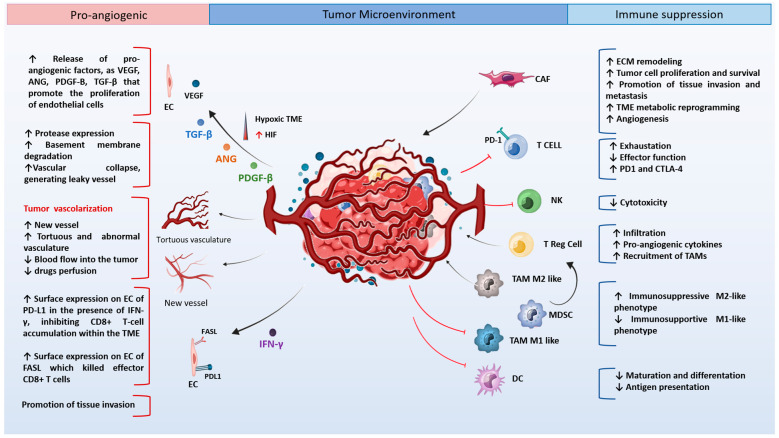
Crosstalk between tumor vasculature and immune cells in the tumor microenvironment. The TME plays a key role in tumor growth by influencing stromal and immune cells. Hypoxia TME promotes tumor vascularization by activating HIF-1α, VEGF, and many chemokines as indicated. Hypoxia increases immune suppression via proinflammatory molecules. Immune cells contribute to both the immunosuppressive tumor microenvironment and abnormal vascularization. Figure created with a Biorender program.

**Figure 2 cancers-16-00491-f002:**
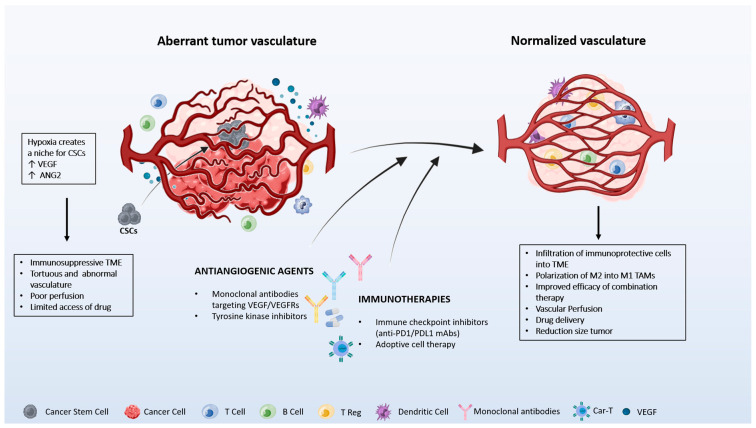
Hypoxia in the tumor microenvironment increases the secretion of VEGF, leading to the generation of leaky and abnormal vasculature limiting the access of drugs to the tumor. Hypoxia also creates a niche for cancer stem cells (CSCs). Antiangiogenic agents normalize the vasculature and create appropriate blood perfusion changes in the immune cell components present in the tumor microenvironment (TME). Tumors with normalized vessels are enriched in tumor-associated macrophages (TAM) with an M1 phenotype, immune protective cells (CD8^+^ T, CD4^+^ T, and dendritic cells), and present low infiltration of immunosuppressive cells (Treg, CCR2^+^ cells). The image is produced using the Biorender program.

**Figure 3 cancers-16-00491-f003:**
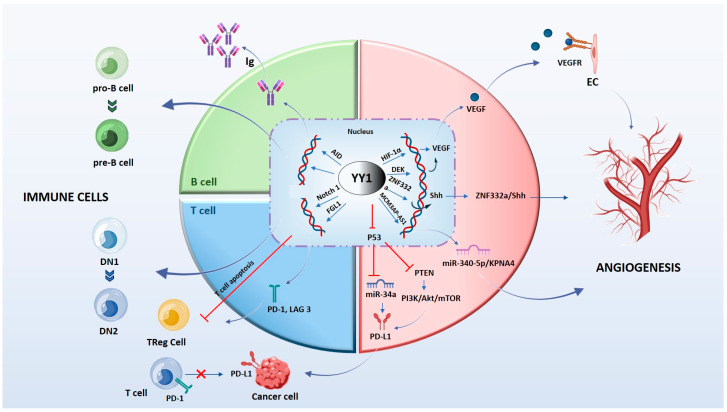
The molecular mechanism by which YY1 regulates immune cells and angiogenesis in the tumor microenvironment. YY1 influences B cells and T cells through several signaling pathways. YY1 is essential for all stages of B cell development and controls processes responsible for antibody maturation and expression of different immunoglobulin. YY1 inhibits the function of T and Treg cells and favors the expression binding to promoter regions of PD-1 and LAG3. YY1 activates the Notch1 pathway, promoting the DN1-to-DN2 T cell transition. YY1 regulates the expression of crucial proteins associated with tumor neovascularization through the interaction with their promoters. Images were prepared using the Biorender program.

**Table 1 cancers-16-00491-t001:** Studies of CAR T cells in the tumor vasculature.

Study			CAR T Cell Design		Study Data	Ref.
Vessel Target	Tumor Type	Study Type/Model	Ag Recognition; Clone	Construct	End Points	
VEGFR2	Metastatic cancer	Phase I/II	scFv anti-human VEGFR-2; KDR1121	CD8/CD28/4–1-BB/CD3ζ	No response/Progression Disease	[123]
VEGFR2	Melanoma (B16F10)	Syngeneic	scFv anti-mouse VEGFR-2; DC101	CD8/CD28/4–1-BB/CD3ζ	Inhibition of tumor growth; increased survival	[124]
	Fibrosarcoma (MCA205)					
	Colon (MC38)					
	Colon (CT26)					
	kidney (RENCA)					
VEGFR1	Lung carcinoma (A549)	Xenograft	scFv anti-human VEGFR1; IMC-18F1	IgG-Fc/CD4/CD3ζ	Inhibition of tumor growth; increased survival; metastasis inhibition	[125]
VEGFR2/3	Breast (MDA-MB-231)	Xenograft	N-terminus VEGF-C	CD8/CD28/CD3ζ	Inhibition of tumor growth; metastasis inhibition	[126]
	Breast (HCC1806)					
TEM8	Breast (MDA-MB-468)	Xenograft	scFv anti-human TEM8; L2	IgG-Fc/CD28/4–1-BB/CD3ζ	Inhibition of tumor growth; increased survival; vascular disruption	[127]
	Breast (LM231)					
	Breast (BCM-2665)	Patient-derived xenograft				
	Breast (WHIM12)					
TEM1	Ewing sarcoma (A673)	Xenograft	scFv anti-human and anti-mouse TEM1; L1C1m	Trilobite engager (CD3/TEM1)	Inhibition of tumor growth	[128]
CLEC14a	Healthy mice	N/A Transgenic	scFV anti-mouse and anti-human CLEC14a; CRT3, CRT5	CD28/CD3ζ	Inhibition of tumor growth; increased survival	[129]
	RipTag2					
	Pancreas (mPDAC)	Syngeneic				
	Lung (LLC)	Syngeneic				
ED-B	Glioma (U87)	Xenograft	scFv anti-human ED-B; L19	CD28/CD3ζ	Inhibition of tumor growth; increased survival	[130]
	Lung (A549)					
	Ewing sarcoma (A673)					
ED-B	Melanoma (B16F10)	Syngeneic	VHH; NJB2; camelid	CD8/CD28/CD3ζ	Inhibition of tumor growth; increased survival	[131]
	Colon (MC38)					
Integrin αvβ3	Melanoma (A375)	Xenograft	scFv anti-human αvβ3; LM609	CD28/CD3ζ	Tumor regression; increased survival	[132]
Integrin αvβ3	Melanoma (B16F10)	Syngeneic	Echistatin (Disintegrin in snake venom)	CD28/CD3ζ	Inhibition of tumor growth; vascular disruption	[133]

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
