# Peer review of "Clinical Potential of YY1-Hypoxia Axis for Vascular Normalization and to Improve Immunotherapy"

_cancers, 2024, doi:10.3390/cancers16030491_

Round 1
Reviewer 1 Report
Comments and Suggestions for Authors
Comments on the current review manusccript on the clinical potential of the H1-Hypoxia axis to improve vascular normalization and immunotherapy and some suggestions that may help to improve it are presented below.
In Figure 1, VEG is written instead of VEGF, one of the proangiogenic factors, which needs to be corrected.
In Figure 2, the statement "rRduction size tumor" from the "normalized vasculature" explanations needs to be corrected.
It is recommended to review the phrase "leaky torturous" in the legend of figure 2.
In the text of the manusccript, many abbreviations are used for which explanations are not written, and it would be appropriate to include their explanations in the text.
Author Response
Reviewer 1
Comments on the current review manusccript on the clinical potential of the H1-Hypoxia axis to improve vascular normalization and immunotherapy and some suggestions that may help to improve it are presented below.
Thanks to reviewer for helpful comments
In Figure 1, VEG is written instead of VEGF, one of the proangiogenic factors, which needs to be corrected.
Many thanks we corrected VEGF in Figure 1
In Figure 2, the statement "rRduction size tumor" from the "normalized vasculature" explanations needs to be corrected.
We corrected the word
It is recommended to review the phrase "leaky torturous" in the legend of figure 2.
We corrected the phrase
In the text of the manusccript, many abbreviations are used for which explanations are not written, and it would be appropriate to include their explanations in the text.
We corrected through the text and written in full the abbreviation when were cited on the first time
Reviewer 2 Report
Comments and Suggestions for Authors
In the present manuscript, the authors describe the interplay between tumor-angiogenesis and immune modulation and focus on the function and mechanisms of the YY1-HIF axis that regulates the vascular and immune tumor microenvironment. The topics covered are well discussed according to previous data.
Author Response
Many Thanks to reviewer comments
Reviewer 3 Report
Comments and Suggestions for Authors
The review "Clinical potential of YY1-hypoxia axis for vascular normalization and to improve immunotherapy" by Meo and de Nigris explores the role of YY1 in regulating the tumor immune microenvironment and angiogenesis. The topic is interesting and relevant in the field and authors give an interesting summary of YY1 role in these processes, including its molecular mechanisms. The review is well-written and engaging. Furthermore, the review can be further improved particularly in language editing to address grammatical errors, as well as some content revisions, as outlined in the forthcoming point-by-point comments.
1. Lane 46-47: In the sentence "In particular, M2-like macrophages secrete pro-angiogenic factors, mainly VEGF, contributing to the development of redundant and immature vessels.", the use of the term "redundant" here might be unclear. To improve clarity, the authors could provide more context or specify whether "redundant" is intended to mean vessels that are unnecessary, excessive, or exhibiting duplication.
2. Lane 59-61: similar to previous comment, in the sentence that states: "However, data emerging from trials revealed that several regulatory mechanisms can occur within the hypoxic tumor microenvironment, resulting in redundant levels of immune suppression and vasculature heterogeneity independent of VEGF", the term "redundant" might be ambiguous. In here, the authors could provide more clarity on whether "redundant" is meant to convey reduced immune suppression and vasculature heterogeneity.
3. Lane 26-34: There is problem with contradicting statement in the discussion regarding the impact of a hypoxic microenvironment on tumor vasculature and cell proliferation. In the paragraph discussing the role of a hypoxic microenvironment in solid tumors, the authors state that it is a pathogenic condition promoting new vessel formation and supporting cancer cell proliferation. However, the authors also state in a subsequent statement describing the vasculature within tumors as heterogeneous, abnormal, and dysfunctional. This contradiction could lead to confusion regarding how the described vasculature characteristics can actually contribute to tumor growth. Could the authors please provide further elaboration on how the described abnormal and dysfunctional vasculature can still facilitate cancer cell proliferation within the context of a hypoxic microenvironment?
4. Lane 41-46: There are some grammatical issues. Please clarify on the use of the phrase "a hypoxic TME increases…". In a biological context, it seems unusual to attribute the action of increasing chemokine expression directly to the hypoxic TME itself, as it is not an active agent that produces chemokine. It would be helpful for the authors to specify the cells or components within the hypoxic TME responsible for the observed increase in chemokine expression. Moreover, the sentence "These cells which contribute to immunosuppression and abnormal vessel growth.” might need rewriting to enhance clarity and improve the grammatical issue.
5. Lane 117: Please provide the full name of abbreviation for FGF2 and PDGFB upon its first mention.
6. Lane 141-143: The transition from the regulation of HIF-1α by YY1 to the negative regulation of p53 by YY1 appears somewhat abrupt. The sudden introduction of p53 regulation disrupts the coherence of the narrative flow in this section. Authors could consider refining the transition or providing a brief contextual bridge to enhance the overall clarity and connectivity of the information presented.
7. Lane 186: The term "miRNA regulatory loops" is used to describe the regulatory relationships here, however, considering the involvement of various non-coding RNAs beyond miRNAs in the described interactions, it might be more accurate to use a term that captures the broader scope of these regulatory networks.
8. Lane 209: Please provide the full name of abbreviation for TNFα.
9. Lane 343-352: In the explanation about the concept of targeting the immune microenvironment as an alternative approach to improving immune responses, there appears to be ambiguity regarding the intended targets. The statement "target the immune microenvironment" (lane 343) might benefit from additional clarification, especially in the example of CAR T-cell therapy. Given that CAR T-cells are commonly engineered to recognize and kill tumor cells expressing tumor-associated antigens or tumor-specific antigens, it would be helpful for the authors to provide more explicit details on whether the CARs, especially those designed to target VEGFR2 as mentioned in Table 1, aim to target tumor cells overexpressing VEGFR2, endothelial cells, or both, or maybe other cells in the tumor immune microenvironment. Furthermore, the authors could provide more description on the relevance of such approach in terms of vascular normalization.
Comments on the Quality of English LanguageThere are no major issues with the English Language of this manuscript, but minor editing of English Language is needed.
Author Response
Revierw 3
The review "Clinical potential of YY1-hypoxia axis for vascular normalization and to improve immunotherapy" by Meo and de Nigris explores the role of YY1 in regulating the tumor immune microenvironment and angiogenesis. The topic is interesting and relevant in the field and authors give an interesting summary of YY1 role in these processes, including its molecular mechanisms. The review is well-written and engaging. Furthermore, the review can be further improved particularly in language editing to address grammatical errors, as well as some content revisions, as outlined in the forthcoming point-by-point comments.
Many thanks for helpful comments we corrected as follow:
1.Lane 46-47: In the sentence "In particular, M2-like macrophages secrete pro-angiogenic factors, mainly VEGF, contributing to the development of redundant and immature vessels.", the use of the term "redundant" here might be unclear. To improve clarity, the authors could provide more context or specify whether "redundant" is intended to mean vessels that are unnecessary, excessive, or exhibiting duplication.
Thanks we changed the word with excessive
2.Lane 59-61: similar to previous comment, in the sentence that states: "However, data emerging from trials revealed that several regulatory mechanisms can occur within the hypoxic tumor microenvironment, resulting in redundant levels of immune suppression and vasculature heterogeneity independent of VEGF", the term "redundant" might be ambiguous. In here, the authors could provide more clarity on whether "redundant" is meant to convey reduced immune suppression and vasculature heterogeneity.
We changed with multiple mechanisms of immune suppression
3.Lane 26-34: There is problem with contradicting statement in the discussion regarding the impact of a hypoxic microenvironment on tumor vasculature and cell proliferation. In the paragraph discussing the role of a hypoxic microenvironment in solid tumors, the authors state that it is a pathogenic condition promoting new vessel formation and supporting cancer cell proliferation. However, the authors also state in a subsequent statement describing the vasculature within tumors as heterogeneous, abnormal, and dysfunctional. This contradiction could lead to confusion regarding how the described vasculature characteristics can actually contribute to tumor growth. Could the authors please provide further elaboration on how the described abnormal and dysfunctional vasculature can still facilitate cancer cell proliferation within the context of a hypoxic microenvironment?
Accordingly we change the phrase as follow “This condition determined insufficient nutrient and oxygen perfusion in selected tumor areas, the core far from vessel, or in high proliferation cohort of cells,. Under reduced oxygen tensions tumor cells to adapt to hypoxia microenvironment activate several pathways to survival among them hypoxia factor 1 (HIF-1) and Vascular endothelial growth factor (VEGF) able to generate a vicious cycle with immune cells.
4.Lane 41-46: There are some grammatical issues. Please clarify on the use of the phrase "a hypoxic TME increases…". In a biological context, it seems unusual to attribute the action of increasing chemokine expression directly to the hypoxic TME itself, as it is not an active agent that produces chemokine. It would be helpful for the authors to specify the cells or components within the hypoxic TME responsible for the observed increase in chemokine expression. Moreover, the sentence "These cells which contribute to immunosuppression and abnormal vessel growth.” might need rewriting to enhance clarity and improve the grammatical issue.
We modify accordingly. “Moreover, cell components of hypoxic tumor microenvironment (TME) increase in chemokines expression, including stromal cell-derived factor 1- α (SDF1-α) and C-C Motif Chemokine Ligand 28 (CCL28), able to recruit regulatory T (Tregs), heterogeneous population of immature myeloid cells (MDSCs), and M2-type macrophages. These immune cells contribute to immunosuppressive microenvironment and promote abnormal vessel growth.
- Lane 117: Please provide the full name of abbreviation for FGF2 and PDGFB upon its first mention.
We added
- Lane 141-143: The transition from the regulation of HIF-1α by YY1 to the negative regulation of p53 by YY1 appears somewhat abrupt. The sudden introduction of p53 regulation disrupts the coherence of the narrative flow in this section. Authors could consider refining the transition or providing a brief contextual bridge to enhance the overall clarity and connectivity of the information presented.
We corrected as follow it inhibits the activation of tumor suppressor p53-
- Lane 186: The term "miRNA regulatory loops" is used to describe the regulatory relationships here, however, considering the involvement of various non-coding RNAs beyond miRNAs in the described interactions, it might be more accurate to use a term that captures the broader scope of these regulatory networks.
We clarify the phrase added “noncoding RNA regulatory mechanism”
- Lane 209: Please provide the full name of abbreviation for TNFα.
We modified
- Lane 343-352: In the explanation about the concept of targeting the immune microenvironment as an alternative approach to improving immune responses, there appears to be ambiguity regarding the intended targets. The statement "target the immune microenvironment" (lane 343) might benefit from additional clarification, especially in the example of CAR T-cell therapy. Given that CAR T-cells are commonly engineered to recognize and kill tumor cells expressing tumor-associated antigens or tumor-specific antigens, it would be helpful for the authors to provide more explicit details on whether the CARs, especially those designed to target VEGFR2 as mentioned in Table 1, aim to target tumor cells overexpressing VEGFR2, endothelial cells, or both, or maybe other cells in the tumor immune microenvironment. Furthermore, the authors could provide more description on the relevance of such approach in terms of vascular normalization.
Accordingly we modified “Recent advances were achieved by CAR T-cell therapy [118]. This approach involves: 1- the molecular modification of human T cells, introducing synthetic receptors, capable to recognize selected antigens as VEGF receptor 2 or tumor-associated antigens (TAAs), 2- autologous infusion of CAR-T cells in patient where they can recognize and kill tumor microenvironment cells expressing selected antigens.

Reviewer 4 Report
Comments and Suggestions for Authors
This is an excellent review addressing the HIF-YY1 axis as a target to improve the efficacy of anti-cancer immunotherapy approaches. The Authors provide a clear overview of the complex tumor microenvironment where a plethora of factors either up- or down-regulate tumor growth through unpredictable, redundant, and overlapping mechanisms of vascularization and immune suppression. Well-prepared figures and tables help the reader across this difficult argument. I have only two concerns that may help to improve the quality of this article further.
For a more balanced perspective, subchapter 3.1 should report the failure of several studies aimed at controlling HIF expression to reduce cancer growth. Note also that hyperoxia (i.e., excess O2) could reduce tumor growth while paradoxically inducing higher HIF expression, at least in prostate cancer. As a matter of fact, the next subchapter introduces the idea that HIF may not be required for cancer growth.
The role of NO in improving the efficacy of immunotherapy by interacting with YY1 has not been developed enough. Adding a few sentences describing the underlying molecular mechanisms and outcomes would be useful.
Minor
Syntax problems in lines 81-81.
Line 86-87, please reword that sentence
Line 189, clarify the meaning of the term “sponging”.
Comments on the Quality of English LanguageOnly very minor issues detected.
Author Response
Reviewer 4
This is an excellent review addressing the HIF-YY1 axis as a target to improve the efficacy of anti-cancer immunotherapy approaches. The Authors provide a clear overview of the complex tumor microenvironment where a plethora of factors either up- or down-regulate tumor growth through unpredictable, redundant, and overlapping mechanisms of vascularization and immune suppression. Well-prepared figures and tables help the reader across this difficult argument. I have only two concerns that may help to improve the quality of this article further.
We thanks the reviewer helpful comments and addressed point by point as follow
For a more balanced perspective, subchapter 3.1 should report the failure of several studies aimed at controlling HIF expression to reduce cancer growth. Note also that hyperoxia (i.e., excess O2) could reduce tumor growth while paradoxically inducing higher HIF expression, at least in prostate cancer. As a matter of fact, the next subchapter introduces the idea that HIF may not be required for cancer growth.
We added this phrase .
.Hypoxia is an integral part of TME and HIF-1 pathways play a central role in tumor growth affecting multiple interconnected mechanisms rendering very challenge target HIF-1. In this context given the central role of YY1 in the stability of HIF-1 on the VEGF promoter, it is conceivable that YY1 targeting might offered innovative therapeutic approach to normalized vascularization
The role of NO in improving the efficacy of immunotherapy by interacting with YY1 has not been developed enough. Adding a few sentences describing the underlying molecular mechanisms and outcomes would be useful.
We added the phrase Additionally NO donor could be useful to improve the efficacy of immunotherapy by modulating YY1activity, because functional inactivation of YY1 could reduce T-reg proliferation and accordingly modify suppressive microenvironment
Minor
Syntax problems in lines 81-81.
We change the sentence as follow From these observations emerged the needed of novel therapeutic regimes to normalize the tumor vascular blood supply and to immune modulate the TME permitting the infiltration of activated immune effectors cells [17,18].
Line 86-87, please reword that sentence
We changed as “treatment administered concomitantly with anti-angiogenic drugs might be more effective than single [13,17,19]. Indeed target both immune-microenvironment…. .
Line 189, clarify the meaning of the term “sponging”.
We change the phrase In some tumors, e.g. acute leukemia, YY1 acting as transcription factor promoted the overexpression of Small nucleolar RNA host gene 5 (SNHG5) that complements and binds miR-26b reducing its proangiogenic modulation
